# Dissipative Photochemical Abiogenesis of the Purines

**DOI:** 10.3390/e24081027

**Published:** 2022-07-26

**Authors:** Claudeth Hernández, Karo Michaelian

**Affiliations:** 1Department of Physics, Division of Exact and Natural Sciences, Campus Hermosillo, Universidad de Sonora, Hermosillo C.P. 83067, Mexico; claudeth.clarissa@gmail.com; 2Department of Nuclear Physics and Application of Radiation, Instituto de Física, Universidad Nacional Autónoma de México, Circuito Interior de la Investigación Científica, Cuidad Universitaria, Cuidad de México C.P. 04510, Mexico

**Keywords:** origin of life, dissipative structuring, non-equilibrium thermodynamics, prebiotic chemistry, abiogenesis, adenine, guanine, hypoxanthine, xanthine, purines, hydrogen cyanide, cyanogen, 05.70.Ln, 91.62.Uv, 91.62.+g, 91.62.Kt, 91.62.Np, 91.70.h, 87.14.gk, 87.14.gn, 87.15.-v, 87.16.Dg, 87.23.-n, 92C05, 92C15, 92C40, 92C45, 92D15, 80A99, 82C99

## Abstract

We have proposed that the abiogenesis of life around the beginning of the Archean may have been an example of “spontaneous” microscopic dissipative structuring of UV-C pigments under the prevailing surface ultraviolet solar spectrum. The thermodynamic function of these Archean pigments (the “fundamental molecules of life”), as for the visible pigments of today, was to dissipate the incident solar light into heat. We have previously described the non-equilibrium thermodynamics and the photochemical mechanisms which may have been involved in the dissipative structuring of the purines adenine and hypoxanthine from the common precursor molecules of hydrogen cyanide and water under this UV light. In this article, we extend our analysis to include the production of the other two important purines, guanine and xanthine. The photochemical reactions are presumed to occur within a fatty acid vesicle floating on a hot (∼80 °C) neutral pH ocean surface exposed to the prevailing UV-C light. Reaction–diffusion equations are resolved under different environmental conditions. Significant amounts of adenine (∼$10^{-5}$ M) and guanine (∼$10^{-6}$ M) are obtained within 60 Archean days, starting from realistic concentrations of the precursors hydrogen cyanide and cyanogen (∼$10^{-5}$ M).

## 1. Introduction

Possible chemical or photochemical routes to the production of the purines, including adenine, hypoxanthine, guanine, xanthine, and 2,6-diaminopurine from common precursor molecules have been extensively studied since the first experiments of Miller and Urey in 1953. Sources of free energy for synthesis have included heat gradients, lightning, UV light, and shock waves. Plausible precursor molecules are hydrogen cyanide (HCN) and water, along with the photochemical and hydrolysis products of HCN, which include formamide (H_2_N-CHO), ammonium formate (NH_4_HCO_2_), cyanogen (NCCN) and cyanate (OCN^−^) [1,2,3,4,5,6,7,8,9,10,11,12,13]. These experiments, however, were usually carried out at alkaline pH and unrealistically large HCN concentrations (>0.1 M) because, only under these conditions, polymerization of HCN to cis-DAMN (an intermediate on route to adenine and guanine) prevails over hydrolysis.

From the perspective of the “Thermodynamic Dissipation Theory of the Origin of Life” [14,15,16,17,18,19,20,21,22,23,24,25,26,27,28], descriptions of molecular synthesis alone, however, are insufficient to capture life’s vitality, which includes proliferation, dynamics, and evolution. The vitality of life requires instead a constant dissipation of an imposed thermodynamic potential. Of the above listed free energy sources employed in synthesis experiments, only UV light would have been sufficiently energy dense and continuously available throughout the Archean. Our theory therefore affirms that the “fundamental molecules of life” (those in the three domains of life) were, at their origin, microscopic photon dissipative structures (i.e., pigments) which arose “spontaneously” under the thermodynamic imperative to dissipate the prevailing UV-C (210–285 nm) solar photon spectrum (Figure 1) [14,15,21,26].

In this paper we model the origin of life as a photochemical dissipative structuring process with reaction–diffusion occurring within a fatty acid vesicle floating at the ocean surface, permeable to the precursor molecules HCN, cyanogen and H_2_O, and under a UV-C photon spectrum arriving at the Earth’s surface during the Archean (Figure 1). The concentrations of the relevant product molecules are determined as a function of ocean surface temperature, and as a function of incident light intensity, varying the amplitude but not the shape of the spectrum of Figure 1. Starting from a realistic environmental HCN concentration of $6\times 10^{-5}$ M, we show that adenine and guanine concentrations of $2.4\times 10^{-5}$ M and $7.6\times 10^{-7}$ M respectively, could have been obtained in only 60 Archean days at neutral pH, 80 °C, and under a nominal UV-C flux of 125 W m^−2^ μm^−1^ at 260 nm. These final purine concentrations increase exponentially with temperature (roughly an order of magnitude for each 10 °C increase in surface temperature) and have a linear but complex dependence on light intensity.

## 2. Thermodynamics of Dissipative Structuring

We have identified the long wavelength part of the UV-C region (∼210–285 nm), plus part of the UV-B and UV-A regions (∼310–360 nm), of the solar spectrum as the thermodynamic potential which could have driven the molecular dissipative structuring [21,26], proliferation, and evolution relevant to the origin of life. This light prevailed at Earth’s surface from the Hadean, before the origin of life (somewhere near the beginning of the Archean ∼3.9 Ga), and for perhaps as long as 1400 million years until the formation of an ozone layer at about 2.5 Ga [29,31,32] after natural oxygen sinks (e.g., volcanic reducing gases such as CH${}_{4}$, CO, H_2_S and Fe${}^{+2}$) became overwhelmed by organisms performing oxygenic photosynthesis, possibly fertilized by volcanic phosphorus [33].

The relevance of this particular region of the solar spectrum, corresponding to the Archean atmospheric window of transparency, to the dissipative structuring of the fundamental molecules is that longer wavelengths do not contain sufficient free energy to directly break double covalent carbon bonds, while shorter wavelengths contain enough free energy to destroy carbon based molecules through successive ionization or fragmentation.

A number of empirical evidences support our dissipative structuring conjecture for the fundamental molecules of life. First, the wavelength of maximum absorption of many of these molecules coincide with the predicted window in the Archean atmosphere (Figure 1). Secondly, many of the fundamental molecules of life are endowed with peaked conical intersections [26] giving them broad band absorption and high quantum yield for internal conversion, i.e., extremely rapid (picosecond) dissipation of the photon-induced electronic excitation energy into vibrational energy of molecular atomic coordinates, and finally into the surrounding water solvent [21,34]. The most convincing evidence of all, however, is that many photochemical routes to the synthesis of nucleic acids [35], amino acids [36], fatty acids [23], sugars [37], and other pigments [18] from common and simple precursor molecules have been identified at these wavelengths and the rate of photon dissipation within the Archean window generally increases after each incremental step on route to synthesis, a behavior strongly suggestive of dissipative structuring in the non-linear non-equilibrium regime [21,23,26]. Finally, even small changes (e.g., tautomerizations) or additions to the fundamental molecules destroys these special optical properties.

The photochemistry involved in the non-equilibrium thermodynamics of microscopic dissipative structuring and proliferation of the fundamental molecules, and how this leads to the evolution of molecular concentration profiles with ever greater solar photon dissipation capacity (corresponding to increases in global entropy production) has been described in detail elsewhere (Michaelian [26]). Here we emphasize only those aspects most important to the dissipative structuring of the purines guanine and xanthine.

## 3. The Dissipative Structuring of the Purines

### 3.1. The Model

Cyanogen, HCN, and its hydrolysis product formamide, were first recognized by Eduard Pflüger in 1875 to be the probable precursors of life [38]. The fundamental molecules; nucleic acids [6,35], amino acids [39,40], fatty acids [37], and even simple sugars [41,42] have all been synthesized from these precursors.

HCN is found throughout the cosmos [43,44]. Astronomical observational data indicates that HCN and water can be found together in unexpectedly high abundance in many hot star forming regions [45]. It may be, therefore, that HCN was incorporated into the bodies that formed the Earth and was gradually out-gassed to the surface, as is now suspected for Earth’s water. Cyanide has, in fact, been found in abundance in carbonaceous chondrite meteorites [46]. Furthermore, if Earth’s atmosphere during the Hadean and Archean had been weakly reducing [30] (see below), HCN production on Earth could have resulted from CH${}_{4}$ photolysis by the solar Lyman alpha line (121.6 nm) in the upper atmosphere, leading to CH${}^{*}$ radicals attacking N${}_{2}$, or, UV (145 nm) photolysis of N${}_{2}$ and the resulting radicals attacking CH or CH${}_{2}$ [47,48].

Cyanogen, NCCN or (CN)${}_{2}$, can be generated from HCN either photochemically [49] or thermally [50]. The Lyman-α line on atmospheric HCN will produce, through photolysis, the CN${}^{-}$ radical with high quantum yield (reaction R177 from Table 1 of Zahnle [51]) and this radical gives cyanogen after interaction with a second HCN molecule (reaction R199 from Table 1 of Zahnle). Both HCN and cyanogen have been found in the atmosphere of Titan [48] where they probably derive from similar UV photochemical processes in the upper or middle atmosphere.

In Michaelian [26] we have addressed the issue of the expected low concentration of HCN in the Archean oceans leading to a low polymerization rate of HCN with respect to its rate of hydrolysis. Traditional approaches to the origin of life have “resolved” this issue by invoking eutectic freezing to increase the solute HCN concentration to values sufficient for biasing the reactions towards polymerization [52,53,54]. This low temperature approach, however, reduces significantly all thermal reaction rates. Instead, we have emphasized the existence of an ocean surface microlayer of high organic concentration, $10^{4}$ or $10^{5}$ times greater than that of bulk water (see Michaelian [26] for discussion and references). We also assume the existence of fatty acid vesicles of ∼100 μm diameter which would allow the incident UV-C light, as well as the small precursor molecules, HCN, cyanogen, and H${}_{2}$O, to permeate the vesicle bi-layer wall (Figure 2), while trapping inside the reaction products due to their larger sizes and larger dipole moments (Table 1). This allows the intermediate and product molecules, as well as the heat from UV-C photon dissipation, to accumulate within the vesicle.

The existence of amphipathic fatty acid hydrocarbon chains, which through Gibb’s free energy minimization spontaneously form lipid vesicles at the ocean surface, is a common assumption in origin of life scenarios [66,67]. Deamer and Georgiou [68,69] have discussed in detail the many roles of lipid membranes in life’s origin. In order to maintain vesicle integrity at the high surface temperatures considered here at the beginning of the Archean, ∼80 °C [70,71,72,73,74,75], these fatty acids would necessarily have been long (∼18 C atoms) and cross linked through UV-C light, which improves stability at high temperatures and over a wider range of pH values [23,76], and provides resistance to salt flocculation. In the anoxic Archean ocean surface environment, UV-C light would promote cross-linking between consecutive fatty acids rather than cause oxidative damage.

Archean fatty acids may have been aromatic, giving them improved stability at higher temperature for shorter lengths due to a van der Waals stacking interaction between the rings of the interior and exterior layers of the bilayer vesicle. There is, in fact, a predominance of 16 and 18 carbon atom fatty acids in the whole Precambrian fossil record [77,78]. Today, the aromatic fatty acid 10-phenyltridecanoic acid (16 carbon atoms) and other ω-phenyl fatty acids, such as 12-phenyldodecanoic acid (18 carbon atoms) and 14-phenyltetradecanoic acid (20 carbon atoms), are found in the highly salt resistant marine bacteria *V. alginolyticus* [79], and ω-cyclohexyl fatty acids are found today in some acidophilic-thermophilic bacteria isolated from hot springs [80].

From the perspective of our dissipative structuring scenario, aromatic fatty acids are also interesting because the conjugated aromatic rings have wide band UV absorption cross section within the 210–285 nm region and a conical intersection for rapid internal conversion to the ground state, giving them large photon dissipation efficacy and photochemical stability under this UV-C light. Indeed, we have suggested that fatty acids, just like the nucleic acids and aromatic amino acids, could have been dissipatively structured from HCN and CO${}_{2}$ saturated water under this UV-C light at moderate temperatures on the ocean surface [23] because these optical characteristics are the hallmarks of dissipative structuring [21,26].

### 3.2. The Photochemical Route to the Purines

Our proposed photochemical route to the purines from the precursors H${}_{2}$O, HCN and cyanogen is based on the experimental data of Ferris and Orgel [35] and Sanchez et al. [6], as well as the quantum mechanical calculations of Boulanger et al. [60], and is given in Figure 3.

**Figure 3 entropy-24-01027-f003:**
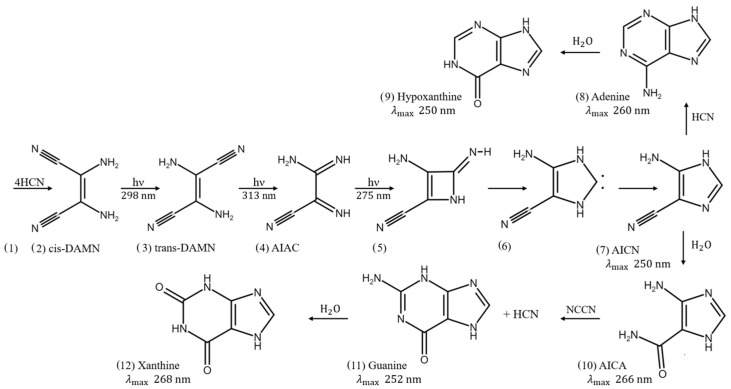
The photochemical synthesis of adenine and guanine from 4 molecules of hydrogen cyanide (HCN) in water, plus a final HCN for adenine (Ferris and Orgel (1966) [35,60]), and a cyanogen molecule for guanine (Sanchez et al. (1968) [6]). Four molecules of HCN are transformed into the smallest stable oligomer (tetramer) of HCN, known as cis-2,3-diaminomaleonitrile (cis-DAMN) (2), which, under a UV-C photon flux isomerizes into trans-DAMN (3) (diaminofumaronitrile, DAFN) which by absorbing two more UV-C photons transforms into an imidazole intermediate, 4-amino-1H-imidazole-5-carbonitrile (AICN) (7). Hot ground state thermal reactions with another HCN molecule, or its hydrolysis product formamide (or ammonium formate), leads to the purine adenine (8). Hydrolysis of adenine (8) leads to hypoxanthine (9). Hydrolysis of AICN (7) leads to AICA (10) and a hot ground state thermal reaction with a cyanogen molecule leads to guanine (11). Hydrolysis of guanine (11) leads to xanthine (12). This is a microscopic dissipative structuring process under UV-C light [21,26] which ends in the purines; adenine, hypoxanthine, guanine and xanthine, all UV-C pigments with large molar extinction coefficients, at 260, 250, 252 and 268 nm respectively, and peaked conical intersections providing rapid dissipation of photons around the wavelength of maximum intensity of the Archean surface solar UV-C spectrum (Figure 1).

The route to guanine results from the hydrolysis of 4-aminoimidazole-5-carbonitrile (AICN) to form 2-amino-3-iminoacrylimidoyl cyanide (AICA) (step (7) to (10) of Figure 3) which then combines through a thermal chemical reaction with cyanogen (NCCN) to form guanine plus a HCN molecule [6]. The reactions on route to adenine and guanine are in competition with hydrolysis and UV lysis, and these relative rates are dependent on concentrations, temperature, pH, the presence of metal ion- and organic intermediate product catalysts, as well as the wavelength dependent intensity of the incident UV spectrum (Section 4). Some of these complexities have been studied by Sanchez et al. [5,6].

In the following subsection we present our simplified out-of-equilibrium kinetic model for 5HCN → adenine and 4HCN + H${}_{2}$O + NCCN → guanine + HCN (Figure 3) photochemical reaction system occurring within a fatty acid vesicle floating within the high organic concentration microlayer of a hot (∼80 °C [70,71,72,73,74,75]) Archean ocean surface under the UV surface spectrum of Figure 1. Favorable comparisons of our simulated photochemical production rates to experimental data have been given in Michaelian [26] (this Special Issue).

### 3.3. The Kinetic Equations

The kinetic equations for the chemical and photochemical reactions are resolved numerically. For this perturbed non-linear reaction–diffusion system it will be shown in Section 4 that various stationary state solutions exist with spatial symmetry breaking resulting in the highest concentration of purines at the center of the vesicle. This would then facilitate a subsequent UV-C polymerization of nucleobases into short oligos (of course assuming the possibility of UV-C-assisted synthesis of ribose from similar precursor molecules [42,81], a UV-C induced nucleoside formation from a base and ribose [82], and a temperature [83] or formamide-catalyzed [84] phosphorylation of the nucleosides [85,86], but these reactions will not be considered here). This stationary state coupling of reactions to diffusion, leading to symmetry breaking (i.e., particular regions of high concentration of the products), was shown to occur for purely thermal reactions with different activator and inhibitor diffusion rates by Turing [87] and studied more generally as *dissipative structures* under the framework of Classical Irreversible Thermodynamic (CIT) theory by Glansdorff and Prigogine [88].

Nomenclature, chemical formula, and abbreviations used throughout the text for the concentrations of the participating chemical species involved in the photochemical reactions leading to the purines shown in Figure 3 are given in Table 1. Their photon extinction coefficients, polar surface area and dipole moments (related to permeability of the vesicle wall), are also given.

From an analysis of published experiment [1,2,3,4,5,6,7,8,9,10,11,12,13,35,55,56,58,89,90,91,92,93,94,95] and time-dependent density functional calculations [60,96], the chemical and photochemical reactions listed in Table 2 will occur in the photochemical dissipative structuring of the purines from HCN, H${}_{2}$O, and NCCN. Below, we describe in detail only those reactions pertinent to the production of guanine and xanthine from AICN. Reactions leading to AICN and adenine have been described in detail elsewhere (Michaelian [26]).

The following is a description of the reactions given in Table 2 by reaction number. A detailed description is given only for those reactions relevant to the production of guanine and xanthine. For a detailed description of all other reactions, see Michaelian [26].

Hydrogen cyanide HCN (H) hydrolyses to formamide H${}_{2}$NCOH (F) [5,91,94];A photon-induced tautomerization coverts formamide (F) into formimidic acid (Fa) [55,56,90,92,93];Formimidic acid (Fa) can, in turn, be photolysed into HCN (H) (or HNC) plus H${}_{2}$O (dehydration) [92,93,97];Formamide (F) hydrolyses to ammonium formate (Af) [93,94];HCN (H) thermally polymerizes into (HCN)${}_{x}$. Its most stable tetramer (x=4) is known as cis-diaminomaleonitrile, cis-DAMN (C) [5];HCN (H) can also thermally polymerize into trans-diaminomaleonitrile, trans-DAMN (T) [5];Trans-DAMN (T) and cyanogen (Cg) are good catalysts for the polymerization of 4HCN into cis-DAMN (see Table 6 of Sanchez et al. [5]). The catalytic effect of trans-DAMN on the tetramization of HCN was incorporated into the model by reducing the energy of the activation barrier such as to give the same amplification factor of 12 due to the catalytic effect of the inclusion of 0.01 M trans-DAMN in the HCN solution observed in the experiments of Sanchez et al. [5] at a temperature of 20 °C.The catalytic effect of cyanogen (Cg) is taken to be the same as that of trans-DAMN (T) since Sanchez et al. determined both of these to be strong catalysts [5]. As mentioned above, cyanogen is a precursor needed for guanine obtained from the Lyman-α line (121.6 nm) on atmospheric HCN giving, through photolysis, the CN${}^{-}$ radical with high quantum yield (reaction R177 Table 1 of Zahnle [51]), and this can form cyanogen (CN)${}_{2}$ by interacting with a second HCN molecule (reaction R199, Table 1 of Zahnle);Trans-DAMN also acts as an auto-catalyst for its own thermal production from 4HCN. Cyanogen is also an effective catalyst for this reaction (Table 6 of Sanchez et al. [5]);(a) By absorbing a photon of 298 nm, cis-DAMN (C) transforms (through rotation around a double carbon covalent bond) into trans-DAMN (T). (b) There is also a smaller quantum efficiency for converting trans-DAMN back into cis-DAMN by absorption of a photon at 313 nm [5,58,60];A photon at 313 nm electronically excites trans-DAMN (T) which then transforms into 2-amino-3-iminoacrylimidoyl cyanide, AIAC (J), through proton transfer from one of the amino groups [60];AIAC (J), on absorbing a photon at 275 nm, transforms through photon-induced cyclicization (ring closure) into an azetene intermediate ((5) of Figure 3) in an electronic excited state, which then transforms to the N-heterocyclic carbene ((6) of Figure 3) and finally this tautomerizes to give the imidazole, 4-aminoimidazole-5-carbonitrle, AICN (I) ((7) of Figure 3) [60];The imidazole AICN (I), created in the previous photochemical reaction #11, is converted through hydrolysis to 4-aminoimidazole-5-carboxamide, AICA (L) [6];Interaction of the imidazole AICN (I) with ammonium formate (Af), together with the catalytic effect of formamide (F), leads to adenine (A) [98,99];Adenine (A) can also be obtained through the attachment of HCN (H) to AICN (I) to form amidine (Am), which is a formamide (F) catalyzed thermal reaction involving formimidic acid (Fa) [100];A subsequent tautomerization of amidine (Am) is required (calculated to have a high barrier of about 50 kcal mol${}^{-1}$) which, once overcome by absorbing a photon at 250 nm, allows the system to proceed through a subsequent barrier-less cyclicization to form adenine (A) [61];Hydrolysis of adenine (A) gives hypoxathine (Hy), determined by Zheng and Meng to have a transition state barrier of 23.4 kcal mol${}^{-1}$ [96];The combination of AIAC (L) with cyanogen (Cg) (a precursor produced in the middle atmosphere from HCN—see reaction # 7) through a thermal reaction leads to guanine (G) and HCN (H) [6];Hydrolysis of guanine (G) leads to xanthine (Xa) [89,95];Photochemical reactions 19 to 28. These represent the absorption (within in a 20 nm region centered on the wavelength of peak absorption) and dissipation through internal conversion at a conical intersection to the ground state on a sub-picosecond time scale. All molecules listed in this set of photo-reactions are photo-stable because of this peaked conical intersection connecting the electronic excited state to the ground state. These reactions, with large quantum efficiencies, represent the bulk of the flow of energy from the incident UV-C spectrum to the emitted outgoing infrared ocean surface spectrum and therefore contribute most to photon dissipation, or entropy production.

To simplify the kinetic equations for the photochemical reactions listed in Table 2, we assume that the molecules only absorb within a region ±10 nm of their maximum absorption wavelength $\lambda_{max}$ and that this absorption is at their maximum molar extinction coefficient $\epsilon_{max}$ (Table 1), and finally that these wavelength regions do not overlap for calculating the shadowing effect of molecular concentrations at higher positions in the vesicle.

We assume that the vesicle is at the ocean surface and the depth coordinate is divided into i=20 bins of width Δx=5
μm and the time step for the recursion calculation for the concentrations is 10 ms. The recursion relation for the factor of light intensity $L_{\lambda }\left(i,C\right)$ for a concentration *C* of the molecule, at a depth x(i)=i·Δx below the ocean surface will be,
(1)$$L_{\lambda }\left(i,C\left(i\right)\right)=L_{\lambda }\left(i-1,C\left(i-1\right)\right)e^{-\Delta x\cdot \alpha_{\lambda }}\cdot 10^{-\Delta x\cdot \epsilon_{\lambda }C\left(i\right)},$$
where $\alpha_{\lambda }$ is the absorption coefficient of water at wavelength λ and $\epsilon_{\lambda }$ is the molar extinction coefficient of the particular absorbing substance which has concentration C(i) at x(i).

The kinetic equations giving the increments in concentration after each time step Δt≡dt, for use in a discrete recursion relation, at a depth *x* below the ocean surface are determined from the reactions listed in Table 2 to be the following:
(2)$$\begin{array}{cc} \frac{dH}{dt} & =D_{H}\frac{\partial^{2}H}{\partial x^{2}}-k_{1}H+d\cdot q_{3}I_{220}L_{220}\left(Fa\right)\frac{\left(1-10^{-\Delta x\epsilon_{220}Fa}\right)}{\Delta x}-k_{5}H^{2}-k_{6}H^{2}-k_{7}H^{2}T-k_{8}H^{2}T+k_{17}LCg\\ \; & =D_{H}\frac{\partial^{2}H}{\partial x^{2}}+d\cdot q_{3}I_{220}L_{220}\left(Fa\right)\frac{\left(1-10^{-\Delta x\epsilon_{220}Fa}\right)}{\Delta x}-Hk_{1}-H^{2}\left(k_{5}+k_{6}+T\left(k_{7}+k_{8}\right)\right)+k_{17}LCg \end{array}$$
(3)$$\begin{array}{cc} \frac{dCg}{dt} & =D_{Cg}\frac{\partial^{2}Cg}{\partial x^{2}}-k_{17}LCg \end{array}$$(4)$$\begin{array}{cc} \frac{dF}{dt} & =D_{F}\frac{\partial^{2}F}{\partial x^{2}}+k_{1}H-d\cdot q_{2}I_{220}L_{220}\left(F\right)\frac{\left(1-10^{-\Delta x\epsilon_{220}F}\right)}{\Delta x}-k_{4}F-k_{14}IFa \end{array}$$(5)$$\begin{array}{cc} \frac{dFa}{dt} & =D_{Fa}\frac{\partial^{2}Fa}{\partial x^{2}}+d\cdot q_{2}I_{220}L_{220}\left(F\right)\frac{\left(1-10^{-\Delta x\epsilon_{220}F}\right)}{\Delta x}-d\cdot q_{3}I_{220}L_{220}\left(Fa\right)\frac{\left(1-10^{-\Delta x\epsilon_{220}Fa}\right)}{\Delta x} \end{array}$$(6)$$\begin{array}{cc} \frac{dAf}{dt} & =D_{Af}\frac{\partial^{2}Af}{\partial x^{2}}+k_{4}F-k_{13}IAf \end{array}$$(7)$$\begin{array}{cc} \frac{dC}{dt} & =D_{C}\frac{\partial^{2}C}{\partial x^{2}}+k_{5}H^{2}+k_{7}H^{2}\left(T+Cg\right)-d\cdot q_{9}I_{298}L_{298}\left(C\right)\frac{\left(1-10^{-\Delta x\epsilon_{298}C}\right)}{\Delta x}\\ \; & \;\;\;+d\cdot q_{9r}I_{313}L_{313}\left(T\right)\frac{\left(1-10^{-\Delta x\epsilon_{313}T}\right)}{\Delta x} \end{array}$$(8)$$\begin{array}{cc} \frac{dT}{dt} & =D_{T}\frac{\partial^{2}T}{\partial x^{2}}+k_{6}H^{2}+k_{8}H^{2}\left(T+Cg\right)+d\cdot q_{9}I_{298}L_{298}\left(C\right)\frac{\left(1-10^{-\Delta x\epsilon_{298}C}\right)}{\Delta x}\\ \; & \;\;\;-d\cdot q_{10}I_{313}L_{313}\left(T\right)\frac{\left(1-10^{-\Delta x\epsilon_{313}T}\right)}{\Delta x}-d\cdot q_{9r}I_{313}L_{313}\left(T\right)\frac{\left(1-10^{-\Delta x\epsilon_{313}T}\right)}{\Delta x} \end{array}$$(9)$$\begin{array}{cc} \frac{dJ}{dt} & =D_{J}\frac{\partial^{2}J}{\partial x^{2}}+d\cdot q_{10}I_{313}L_{313}\left(T\right)\frac{\left(1-10^{-\Delta x\epsilon_{313}T}\right)}{\Delta x}-d\cdot q_{11}I_{275}L_{275}\left(J\right)\frac{\left(1-10^{-\Delta x\epsilon_{275}J}\right)}{\Delta x} \end{array}$$(10)$$\begin{array}{cc} \frac{dI}{dt} & =D_{I}\frac{\partial^{2}I}{\partial x^{2}}+d\cdot q_{11}I_{275}L_{275}\left(J\right)\frac{\left(1-10^{-\Delta x\epsilon_{275}J}\right)}{\Delta x}-k_{12}I-k_{13}IAf-k_{14}IFa \end{array}$$(11)$$\begin{array}{cc} \frac{dL}{dt} & =D_{L}\frac{\partial^{2}L}{\partial x^{2}}+k_{12}I-k_{17}LCg \end{array}$$(12)$$\begin{array}{cc} \frac{dAm}{dt} & =D_{Am}\frac{\partial^{2}Am}{\partial x^{2}}+k_{14}IFa-d\cdot q_{15}I_{250}L_{250}\left(Am\right)\frac{\left(1-10^{-\Delta x\epsilon_{250}Am}\right)}{\Delta x} \end{array}$$(13)$$\begin{array}{cc} \frac{dA}{dt} & =D_{A}\frac{\partial^{2}A}{\partial x^{2}}+d\cdot q_{15}I_{250}L_{250}\left(Am\right)\frac{\left(1-10^{-\Delta x\epsilon_{250}Am}\right)}{\Delta x}+k_{13}IAf-k_{16}A \end{array}$$(14)$$\begin{array}{cc} \frac{dHy}{dt} & =D_{Hy}\frac{\partial^{2}Hy}{\partial x^{2}}+k_{16}A \end{array}$$(15)$$\begin{array}{cc} \frac{dG}{dt} & =D_{G}\frac{\partial^{2}G}{\partial x^{2}}+k_{17}LCg-k_{18}G \end{array}$$(16)$$\begin{array}{cc} \frac{dXa}{dt} & =D_{Xa}\frac{\partial^{2}Xa}{\partial x^{2}}+k_{18}G, \end{array}$$
where the differentials are calculated discretely (e.g., dH/dt≡ΔH/Δt) and all concentrations are calculated at discrete time steps of Δt=10 ms and the calculated value of the change (e.g., ΔH(j)/Δt) for time step *j* is summed to the previous value (e.g., H(j−1)). The day/night factor *d* is equal to 1 during the day and 0 at night. $I_{220},I_{298},I_{313},I_{275}$ and $I_{250}$ are the intensities of the photon fluxes at 220,298,313,275 and 250 nm respectively (Figure 1). $\epsilon_{\lambda }$ are the coefficients of molar extinction for the relevant molecule at the corresponding photon wavelengths λ.

### 3.4. Vesicle Permeability and Internal Diffusion

The permeability of the fatty acid vesicle will decrease both with increasing size of the molecule and with increasing size of its electric dipole moment, and increase with temperature. The final and intermediate product molecules have large sizes and large dipole moments compared with the precursors, implying a tendency towards entrapment within the vesicle. We assume that the vesicle is completely permeable to H${}_{2}$O, HCN (H) formimidic acid (Fa) and cyanogen (Cg) but impermeable to all the other intermediate and final products. Note that ammonium formate (produced within the vesicle in reaction #4) would be in its ionic form and therefore also unable to cross the fatty acid membrane (permeability across lipid membranes are reduced by orders of magnitude if the molecules are polar or charged [101]).

The diffusion constant D${}_{Y}$ for the molecule *Y* within the inner aqueous region of the vesicle will depend on the interior solution viscosity, which is dependent on the concentration of organic material. Studies of inter-cellular diffusion of nucleotides indicate three factors influencing diffusion rates, besides temperature, at high solute densities; the viscosity of the solvent, collision rate dependent on solute concentration, the size of the molecules, and the binding interactions between molecules [102]. The diffusion constant of adenine in pure water has been determined to be $D_{A}=7.2\times 10^{-6}$ cm${}^{2}$ s${}^{-1}$ [103] while the measured diffusion rates in the cytoplasm of a cell can be as low as $1.36\times 10^{-6}$ cm${}^{2}$ s${}^{-1}$ [102].

Films of organics and trace metals, with a high density of lipids and other hydrocarbons, produced, for example, by the ultraviolet spectrum of Figure 1 on CO${}_{2}$ saturated water [23], would be expected on the ocean surface during the Archean. Diffusion constants within this sea surface microlayer would then be significantly smaller than for bulk water. Diffusion rates inside the vesicle will depend on the amount of organic material captured at the air/water interface during the formation of the vesicle (which would have considerable spatial variability) and on the amount of ongoing organic synthesis within the vesicle.

The Stokes-Einstein diffusion constant of uncharged spherical molecules is given by:(17)$$D=\frac{k_{B}T}{6\pi \eta r},$$
where *T* is the absolute temperature, η the dynamic viscosity of the solvent, and *r* the radius of the particle (assumed spherical). For polar solvents such as water, molecules with a charge or dipole moment experience further drag due to the charge-dipole or dipole-dipole interaction [104]. All diffusion constants can, therefore, be approximated relative to that measured for adenine at 300 K through the formula:(18)$$D_{Y}=\frac{\left(2+\mu_{A}\right)A_{A}^{1/2}}{\left(2+\mu_{Y}\right)A_{Y}^{1/2}}\frac{T}{300}D_{A}\left(300\right),$$
where $A_{A}$ is the polar surface area of adenine, and $\mu_{A}$ and $\mu_{Y}$ are the dipole momenst of adenine and the molecule *Y* respectively (Table 1). The factor of 2 in the equation weighs the importance of including the dipole moment in the diffusion calculation (Equation (Equation 17)), which is complex but relatively unimportant to final results.

Our analysis is performed with the diffusion constant for adenine found in present day cytoplasm [102] (i.e., $D_{A}\left(300\right)\approx 1\times 10^{-6}$ cm${}^{2}$ s${}^{-1}$). Using the values given in Table 1 for the molecular dipole moments and their polar areas, and Equation (Equation 18), in Table 3 we list the results for the diffusion constants of all relevant molecules at 80 °C (353.15 K) with respect to that for adenine at 300 K, $D_{A}\left(300\right)$.

Cyclical boundary conditions are assumed for diffusion, except for HCN (H), formimidic acid (Fa) and cyanogen (Cg) which, because of their small size and small dipole moment, can permeate the vesicle wall and, therefore, at the wall they are given their fixed value specified in the initial conditions of the environment outside the vesicle (see below). The second order derivatives for calculating the diffusion were obtained using the second order finite difference method with double precision variables.

### 3.5. Initial Conditions

Miyakawa, Cleaves and Miller [91] estimated the steady state bulk ocean concentration of HCN at the origin of life assuming production, through electric discharge on a neutral atmosphere containing some methane, of radicals which attack N${}_{2}$, leading to an input rate to the oceans of 100 nmole cm${}^{-2}$ y${}^{-1}$, and loss of HCN due to hydrolysis and destruction at submarine vents with a 10 million year cycling time of all ocean water for an ocean of 3 km average depth. For an ocean of pH 6.5 and temperature of 80 °C, they obtained a value of [HCN] $=1.0\times 10^{-10}$ M [91].

Not included in the Miyakawa et al. calculation, however, is, as mentioned above, the fact that HCN can also be produced through the solar Lyman alpha line (121.6 nm) photo-lysing N${}_{2}$ in the upper atmosphere giving atomic nitrogen which then combines with CH and CH${}_{2}$ to give HCN, or through 145 nm photolysis of CH${}_{4}$ leading to a CH${}^{*}$ radical which attacks N${}_{2}$ to give HCN [47]. Including this UV production would increase the input of HCN concentration to the oceans by a factor of at least 6 [51,105,106]. More importantly, the first ∼100 μm of the ocean surface is now known to be a unique region (the hydrodynamic boundary layer) where surface tension leads to enriched organics with densities up to $10^{4}$ or $10^{5}$ times that of organic material in the water column slightly below [107]. Langmuir circulation, Eddy currents, and the scavenging action of bubbles, tend to concentrate organic material into this surface film. Trace metal enhancement in this microlayer can be one to three orders of magnitude greater than in the bulk [107,108].

This high density of organic material trapped through hydrophobic and ionic interactions at the ocean surface leads to significantly lower diffusion rates within the surface microlayer as compared to the ocean bulk [107]. Little diffusion and turbulence imply little mixing. The ocean microlayer is, therefore, a very stable layer which would not be recycled through ocean vents and thus allowed to accumulate. Finally, although HCN is very soluble in bulk water, molecular dynamic simulations have shown that it concentrates to about an order of magnitude larger at the air-water interface due to lateral HCN dipole-dipole interactions, and that HCN evaporates at lower rates than water [109].

Therefore, rather than assuming the low HCN bulk concentrations of Miyakawa et al., we instead consider a higher initial surface concentration of HCN (H) and cyanogen (Cg) of $6\times 10^{-5}$ and for formamide (F) and formimidic acid (Fa) of $1\times 10^{-5}$, the latter resulting from a photochemical tautomerization of formamide, the hydrolysis product of HCN (reactions #1 and #2 of Table 2). We also allow for perturbation of the system by considering the probable existence of small and sparse patches of much higher concentration of the precursors, justified by the characteristics of the ocean microlayer and the dipole-dipole interaction between HCN molecules. The system is perturbed by a single patch of relatively high concentration (0.1 M) of HCN, cyanogen (Cg), and formimidic acid (Fa) (see Michaelian [26] for a description) into which our vesicle is assumed to drift (at day 10.4) only once for 2 min during 60 Archean days. The initial concentrations of all other reactants inside the vesicle (assumed impermeable to these) are taken to be $1.0\times 10^{-10}$ M.

Our system is under a diurnal 8 h constant flux of radiation followed by an 8 h period of darkness during which thermal reactions proceed, but not photochemical reactions. Different surface UV-C light intensities will be considered due to the uncertainty in the Archean atmospheric gas concentration profile. The “nominal” intensity corresponds to a weakly reducing and self-shielding (for λ<220 nm) atmosphere as proposed by Sagan and Chyba [30], which assumes that most of the greenhouse warming (required to solve the Faint Young Sun paradox [30]) is provided by CH${}_{4}$ and a small amount of ammonia (mixing ratio $1\times 10^{-5}$) which is shielded by the products resulting from CO${}_{2}$ and CH${}_{4}$ photolysis (e.g., HCN and aldehydes). The corresponding integrated (210–285 nm) UV-C light intensity for this “nominal” scenario (Figure 1) is 4 W m${}^{-2}$. On the other hand, Cnossen et al. [32] suggest a neutral atmosphere in which the partial pressure of CO${}_{2}$ has to be very large, 0.24 bar, in order to achieve the required global warming with the faint young sun. This results in a much reduced integrated (210–285 nm) UV-C surface flux of 0.10 W m${}^{-2}$. We therefore perform our simulations for a number of different incident photon intensities covering both atmospheric profile scenarios.

## 4. Results

### Evolution of the Concentration Profile

In Michaelian [26] we have validated our photochemical reaction–diffusion model by comparing results to experimental data for the time dependence of a number of different intermediate products.

In Figure 4, Figure 5, Figure 6 and Figure 7 we present the concentrations of the relevant molecular products within the vesicle as a function of time in Archean days (16 h) in the dissipative photochemical synthesis of adenine and guanine, obtained by solving simultaneously the differential kinetic Equations (2)–(16), for the initial conditions and diffusion constants listed in the figure captions.

The concentration profiles of the molecules evolve over time because of accumulation of photo-products within the vesicle and due to a deliberate external perturbation of the non-linear system at 10.4 days (0.1 M of HCN, cyanogen, and formimidic acid) which leads it to a new stationary state in which conversion of the environmental precursor molecules HCN and cyanogen into adenine and guanine occurs at greater rates. This leads to greater photon dissipative efficacy of the system, i.e., to a concentration profile of the product molecules which dissipates more efficiently the incident UV-C spectrum.

In Figure 6 we plot the concentration profiles of the photochemical products obtained also at 80 °C but with 1/100 of the incident light flux (i.e., UV-C flux 1.25 W m${}^{-2}$
μm${}^{-1}$ at 260 nm, integrated flux 210–285 nm = 0.040 W m${}^{-2}$). This would correspond approximately to the illumination on Earth’s surface for a very neutral atmosphere if the only greenhouse gas to resolve the Faint Young Sun paradox (and increase the surface temperature to about 80 °C) was CO${}_{2}$ (see Section 3.2), assuming the standard solar model. It is interesting to note that the adenine concentration after 60 days actually increases slightly with this lower surface UV-C light flux, while all other purine concentrations decrease. This is because less UV-C light implies more formamide (F) being available (less of photoreaction #2) and thus more ammonium formate (Af) (reaction #4). Formamide (F) is a catalyst in the most important thermal reaction #13 which has a low activation energy barrier for converting AICN (I) and ammonium formate (Af) into adenine (A).

Our resulting adenine concentrations are between about one and two orders of magnitude greater than those of guanine, which is consistent with experimental data of Levy et al. [10] using ammonium cyanide as a precursor under long time eutectic freezing conditions. This greater difficulty of producing guanine abiotically through both processes, eutectic freezing and photochemical dissipation, may explain why the major energy storage molecule of life is adenine triphosphate (ATP) rather than guanine triphosphate (GTP).

Given the fixed concentrations of HCN (H), cyanogen (Cg) and formimidic acid (Fa) in the environment, to which the vesicle is permeable, photochemical reactions occur during daylight hours (marked as violet colored sections of the horizontal dashed line labeled as D/N). This gives rise to the observable diurnal oscillations in the concentrations of trans-DAMN (T) and AICN (I) since these are direct products of photochemical reactions.

At 10.4 Archean days, the vesicle is perturbed by assuming it passes through a region of high concentration of HCN (H), cyanogen, and formimidic acid (Fa), each of 0.1 M, for a 2 min period (vertical lines at 10.4 days). This sudden impulse in HCN, cyanogen, and formimidic acid concentration gives rise to rapid increases in all concentrations within the vesicle, in particular formamide (F), the hydrolysis product of HCN, which is an important catalyst for reaction #13 which produces adenine (A) from AICN (I) (see Table 2) and this reaction has a low activation barrier energy. Ammonium formate (Af) is used up in this reaction so its concentration decreases rapidly after the perturbation. More importantly, however, immediately after the perturbation there is a greater production of trans-DAMN (T) in the vesicle and since T acts as a catalyst for the polymerization of HCN (H) (reactions #7 and #8), this produces greater metabolism of HCN into DAMN within the vesicle, leading to a stronger diffusion of HCN into the vesicle from the outside environment. In other words, the reason that a short impulse of HCN, cyanogen, and formimidic acid give rise to an important increase in the rate of production of adenine and guanine is that the vesicle’s semi-permeable wall, together with the set of equations describing the photochemical and chemical reactions, Equations (2)–(16), form a non-linear (auto- and cross-catalytic) system with multiple solutions at any given time.

The perturbation at 10.4 days causes the system to leave the attraction basin of one solution determined by its initial conditions and evolve towards a different, and more probable, stationary state with a higher rate of production of adenine and guanine (slope of the black and blue traces respectively of Figure 5). The new stationary state is more stable than the initial state because its concentration profile is more dissipative, i.e., with more molecules having conical intersections dissipating the absorbed photon energy rapidly into heat, and, therefore, there is less probability of photochemical reactions reverting the concentration profile to the initial state. There is thus a thermodynamic driving force for evolution to the new stationary state after the perturbation due to the system obtaining a greater photon dissipative capacity [26]. For systems where local thermodynamic equilibrium (in space and time) can be defined (see Section 3 of Michaelian [26]), this implies that the entropy production of the system increases (see below). This example of the dissipative structuring a UV-C pigment concentration profile, elucidates the physics and chemistry behind a thermodynamic selection relevant to biological evolution at its earliest stages [26].

The temperature dependence of the concentrations of the product molecules obtained after 60 Archean days is given in Figure 8. Ammonium formate (Af) is produced by the hydrolysis of first HCN (H) to formamide (F) (reaction #1) and then hydrolysis of formamide to Af (reaction #4). Both of these reactions have high activation energies and this results in Af only being produced in important quantities at temperatures greater than about 80 °C. Most of the adenine (A) production occurs through reaction #13 which consumes Af and therefore high temperatures are important to the production of adenine. The hydrolysis of AICN (I) to AICA (L) (step (7) to (10) of Figure 3) also has a high activation energy barrier (reaction #12), also implying high temperatures necessary for the production of hypoxanthine (Hy), guanine (G) and xanthine (Xa), as can be seen by the slopes of their traces on Figure 8.

In Figure 9 we plot the concentrations of the product molecules obtained at a temperature of 80 °C after 60 Archean days as a function of the incident light intensity in the 210–285 nm region, by scaling the spectrum given in Figure 1. Maximum adenine concentration is obtained for fluxes of about 1.25 W m${}^{-2}$
μm${}^{-1}$ at 260 nm, however, at this flux, the concentration of guanine is 3 orders of magnitude smaller. At the nominal intensity of 125 W m${}^{-2}$
μm${}^{-1}$ at 260 nm (Figure 1), the guanine concentration is about 30 times lower than that of adenine which remains high at $2.4\times 10^{-5}$ M.

In Figure 10 we plot the entropy production as a function of time in Archean days due solely to the absorption and dissipation of photons into heat through the conical intersections of the product molecules associated with reactions 19 to 28 of Table 2 for the particular concentration profile at the given time. For a description of the entropy production and its evolution through time in relation to the non-equilibrium thermodynamic Glansdorff–Prigogine universal evolutionary criterion, see reference [26].

## 5. Discussion

As Figure 8 illustrates, high temperatures are very important to a building up high concentrations of purines within the vesicle. High temperatures also foment phosphorylation with phosphate salts using formamide as a catalyst, favoring the formation of acyclonucleosides and the phosphorylation and trans-phosphorylation of nucleosides which only occurs efficiently at temperatures above 70 °C [83,84]. High temperatures are similarly important for the abiotic production of amino acids and fatty acids. Such high temperatures are, in fact, consistent with the geochemical data of the early Archean [70,71,72,73,74] and with the genome comparison of strongly conserved ribosomal RNA [75].

Important concentrations of all purines buildup within the vesicle within only about 60 Archean days under the nominal environmental conditions considered here. There is no need to begin with unrealistically large initial concentrations of HCN by invoking eutectic freezing, and there is no need for highly alkaline conditions in order to favor HCN polymerization over hydrolysis.

It is instructive to compare our overall non-equilibrium results obtained with the model of UV-C dissipative structuring of adenine and guanine from HCN within a lipid vesicle at 80 °C with the quasi-equilibrium experiments of Ferris et al. [39] using high HCN concentration, alkaline pH, and room temperature. Starting with a concentration of HCN (0.1 M) in water (pH 9.2), and allowing this solution to polymerize in the dark at room temperature for a 7 months, and then subjecting these polymers to hydrolysis at 110 °C for 24 h, Ferris et al. obtain an adenine yield of 1 mg L${}^{-1}$, equivalent to a concentration of $7.4\times 10^{-6}$ M (the molar mass of adenine being 135.13 g mol${}^{-1}$). Our model gives a similar adenine concentration after only 26 Archean days (Figure 5), starting from a much lower and more realistic initial concentration of HCN of only $6.0\times 10^{-5}$ M (with only one perturbation of 0.1 M for two minutes) and a more probable Archean neutral pH of 7.0 at 80 °C and under a UV-C flux integrated from 210–285 nm of about 4 W m${}^{-2}$ during daylight hours (Figure 5). At 90 °C our adenine concentration within the vesicle surpasses Ferris et al. value after the perturbation at only 10.4 days (Figure 4).

As detailed in Michaelian [26], besides entrapment of product molecules inside the vesicle, another concentration mechanism for these molecules that may arise is the coupling between reaction and diffusion in the non-linear regime which leads to the breaking of spatial symmetry. For low diffusion rates, the homogeneous stationary state is no longer stable with respect to space dependent perturbations and intermediate and final products may become preferentially concentrated and be consumed in the center of the vesicle (see Michaelian [26]).

The dependence of the final concentrations on the light intensity is complex (Figure 9) but important values of both adenine and guanine can be obtained for surface intensities corresponding to the nominal value (for a weakly reducing atmosphere) of 125 W m${}^{-2}$
μm${}^{-1}$ at 260 nm (Figure 1) and for intensities either one order of magnitude smaller or greater. This result may have a dependence on the particular shape of the incident spectrum, but this was not studied here.

Inorganic catalysts have not been included in our reaction scheme. These can increase significantly the rate of purine production. For example Cu${}^{+2}$ ions increase the rate constant for the conversion of HCN (H) to cis-DAMN (C) [5] (reaction # 5). Metal ions would have been in high abundance at the ocean surface microlayer [107,110].

The dissipative structuring of the pyrimidines under the same Archean UV spectrum (Figure 1) will be considered in a future article.

## 6. Conclusions

We have presented a simple model of UV-C photochemical dissipative structuring of the purines within a lipid vesicle permeable to the common precursor molecules H${}_{2}$O, HCN, cyanogen, and formimidic acid, but impermeable to the reaction products, floating on a hot Archean ocean surface. The chemical and photochemical kinetics are based on published experimental and quantum mechanical *ab initio* data. The surface physical conditions are consistent with biochemical and geochemical fossil evidence from the early Archean.

High temperatures are important to obtaining high purine final concentrations which increase by roughly an order of magnitude for each 10 °C increase in surface temperature. Important concentrations of all purines buildup within the vesicle within only about 60 Archean days. There is no need to assume unrealistically high concentrations of HCN by invoking eutectic freezing or highly alkaline conditions. Our resulting adenine concentrations are between 1 and 2 orders of magnitude greater than those of guanine.

The final product concentrations have a complex dependence on light intensity. The largest concentrations of adenine occur for light intensities of between 1 and 125 W m${}^{-2}$
μm${}^{-1}$ at 260 nm, with guanine concentrations reaching about 1/30 of adenine concentration at 125 W m${}^{-2}$
μm${}^{-1}$ at 260 nm (Figure 9). This adenine to guanine ratio is consistent with experimental data of Levy et al. [10] employing long time eutectic freezing conditions. This greater difficulty of producing guanine abiotically, through both eutectic freezing and photochemical dissipation, may explain the fact that the major energy storage molecule of life is adenine triphosphate (ATP) rather than guanine triphosphate (GTP).

Perturbations, caused by the vesicle floating into isolated patches of higher concentration precursors floating on the ocean surface, can provoke the non-linear auto-catalytic system into new states of higher product productivity with a greater “metabolism” of both precursor molecules (HCN and cyanogen) from the environment and the incident photons, leading to a concentration profile of much greater photon dissipative efficacy. This evolution towards stationary states of usually higher photon dissipative efficacy is the essence of biological evolution which continues today, but now involving the whole solar spectrum and the entire biosphere.

Evolution during the Archean was towards concentration profiles of product molecules with an absorption maximum closer to the peak intensity of the incident UV-C spectrum at 260 nm (Figure 1) and with peaked conical intersections to internal conversion, both increasing the overall efficacy of the dissipation of the incident UV-C solar flux.

For low diffusion rates, there can be significant coupling of reactions with diffusion, leading to non-homogeneous distributions of some of the intermediate products, in particular, with greater concentration of the products at the center of the vesicle (see Michaelian [26]). Such spatial symmetry breaking could facilitate further dissipative structuring, such as polymerization of the nucleobases into nucleic acid.

Dissipative structuring, dissipative proliferation, and dissipative selection (as explained in detail in Michaelian [26]) are the necessary and sufficient components for an explanation in physical-chemical terms of the synthesis, proliferation, and evolution of organic molecules on planets, comets, asteroids, and interstellar space [18] and, in particular, for explaining the origin and evolution of the fundamental molecules of life on Earth.

## Figures and Tables

**Figure 1 entropy-24-01027-f001:**
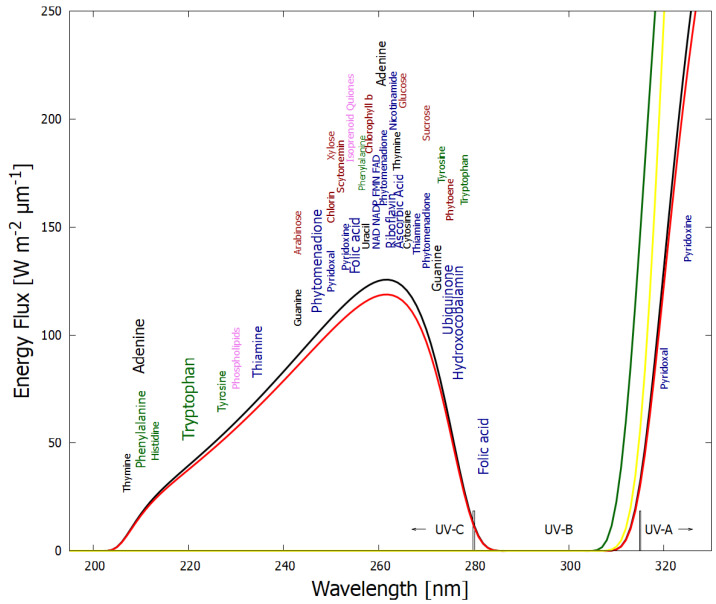
The spectrum of UV-C light available at Earth’s surface before the origin of life at approximately 3.9 Ga and until at least 2.9 Ga (curves black and red respectively) and perhaps throughout the entire Archean until 2.5 Ga. CO${}_{2}$ and probably some H${}_{2}$S were responsible for absorption at wavelengths shorter than ∼205 nm, and atmospheric aldehydes (common photochemical products of CO${}_{2}$ and water) absorbed between about 285 and 310 nm [29], approximately corresponding to the UV-B region. Around 2.2 Ga (green curve), UV-C light at Earth’s surface was completely extinguished by oxygen and ozone resulting from organisms performing oxygenic photosynthesis. The yellow curve corresponds to the present surface spectrum. Energy fluxes are for the sun at the zenith. The names of the fundamental molecules of life are plotted at their wavelengths of maximum absorption; nucleic acids (black), amino acids (green), fatty acids (violet), sugars (brown), vitamins, co-enzymes and cofactors (blue), and pigments (red) (the font size of the letter roughly correspond to the relative size of their molar extinction coefficient). In this work, 125 W m${}^{-2}$
μm${}^{-1}$ is the “nominal” incident intensity at 260 nm, but we also consider lesser and greater intensities due to the uncertainties in the CO${}_{2}$ atmospheric concentration at the origin of life [30]. Adapted from Michaelian and Simeonov [18].

**Figure 2 entropy-24-01027-f002:**
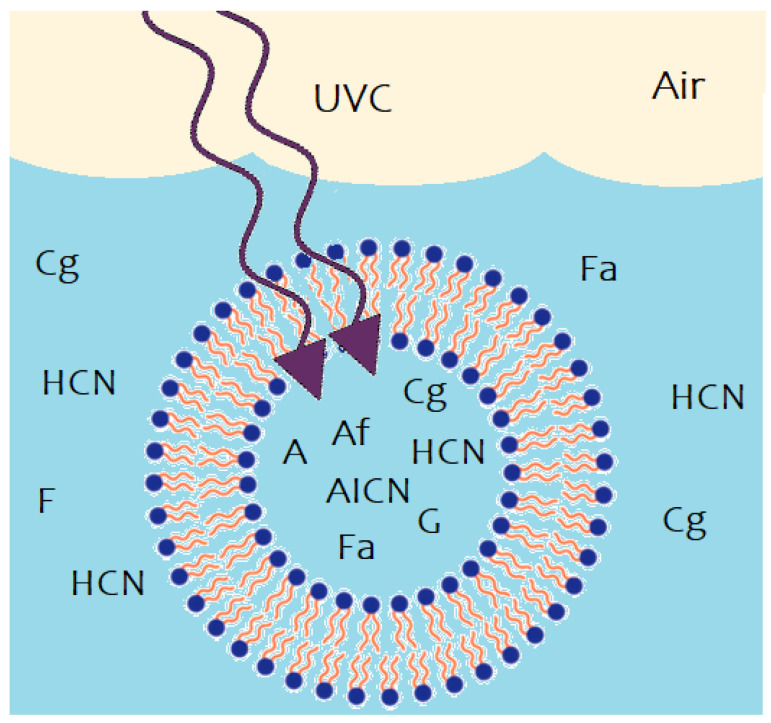
Fatty acid vesicle floating at the ocean surface microlayer of high organic concentration, transparent to UV-C light and permeable to the precursor molecules having small size and dipole moments; H${}_{2}$O, HCN (H), cyanogen (Cg), and formimidic acid (Fa), but impermeable to the intermediate and final photochemical products (e.g., ammonium formate (Af), cis-DAMN, AICN, adenine (A), guanine (G)), which are larger in size and have larger dipole moments (Table 1).

**Figure 4 entropy-24-01027-f004:**
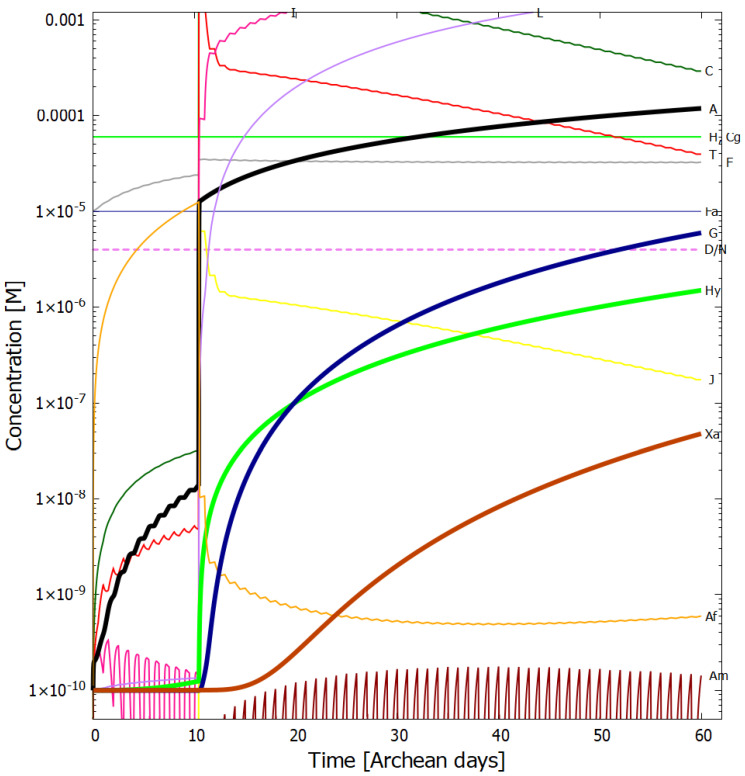
Concentrations as a function of time in Archean days (16 h) of the precursor and product molecules; H—HCN, F—formamide, Fa—formimidic acid, Cg—cyanogen, Af—ammonium formate, C—cis-DAMN, T—trans-DAMN, A—adenine, I—AICN, J—AIAC, L—AICA, Am—amidine, Hy—hypoxanthine, G—guanine, Xa—xanthine, dissipatively structured on route to the synthesis of adenine (black trace) and guanine (blue trace). The initial conditions are; UV-C flux 125 W m${}^{-2}$
μm${}^{-1}$ at 260 nm (Figure 1), integrated flux 210–285 nm = 4 W m${}^{-2}$, temperature T = 90 °C, initial concentrations [H]${}_{0}$ and [Cg]${}_{0}=6.0\times 10^{-5}$ M, [F]${}_{0}$ and [Fa]${}_{0}=1.0\times 10^{-5}$ M and all other initial concentrations [Y]${}_{0}=1.0\times 10^{-10}$ M. The diffusion constant factor $D_{A}\left(300\right)$ was $1.0\times 10^{-6}$ cm${}^{2}$ s${}^{-1}$. There is one perturbation of the system corresponding to the vesicle floating into a region of HCN (H), cyanogen (Cg), and formimidic acid (Fa) of concentration 0.1 M each for two minutes at 10.4 Archean days. A new stationary state at much higher adenine and guanine concentrations is reached after the perturbation. The violet horizontal dashed line labeled “D/N” identifies alternate periods of daylight (violet) and night (blank). After 60 Archean days, the concentration of adenine within the vesicle (black trace) has grown by more than six orders of magnitude, from $1.0\times 10^{-10}$ to $1.19\times 10^{-4}$ M and guanine (blue trace) by five orders of magnitude to $3.20\times 10^{-5}$ M.

**Figure 5 entropy-24-01027-f005:**
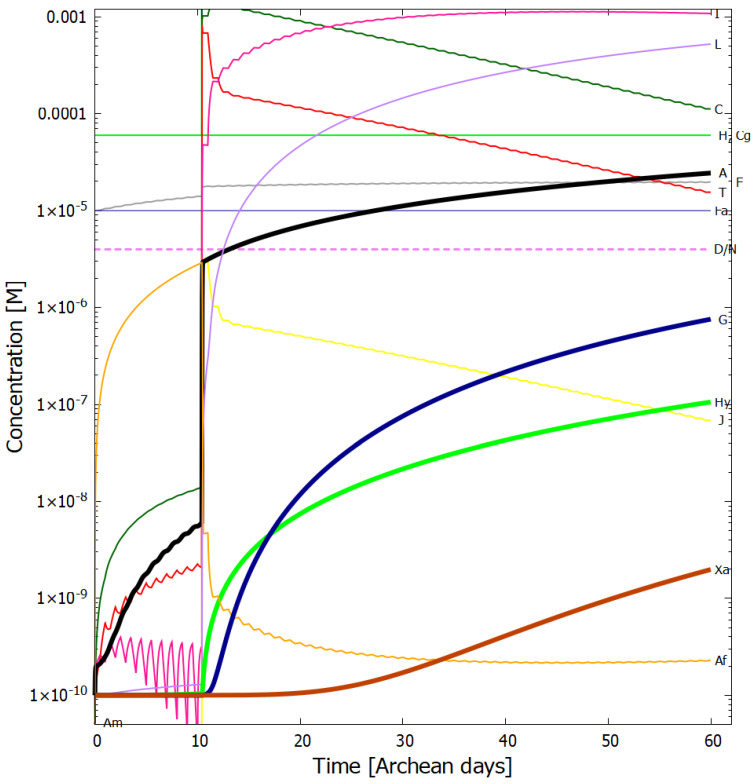
The same as for Figure 4 except for a temperature of 80 °C. The adenine concentration reaches $2.4\times 10^{-5}$ M and a guanine concentration of $7.6\times 10^{-7}$ M after 60 Archean days.

**Figure 6 entropy-24-01027-f006:**
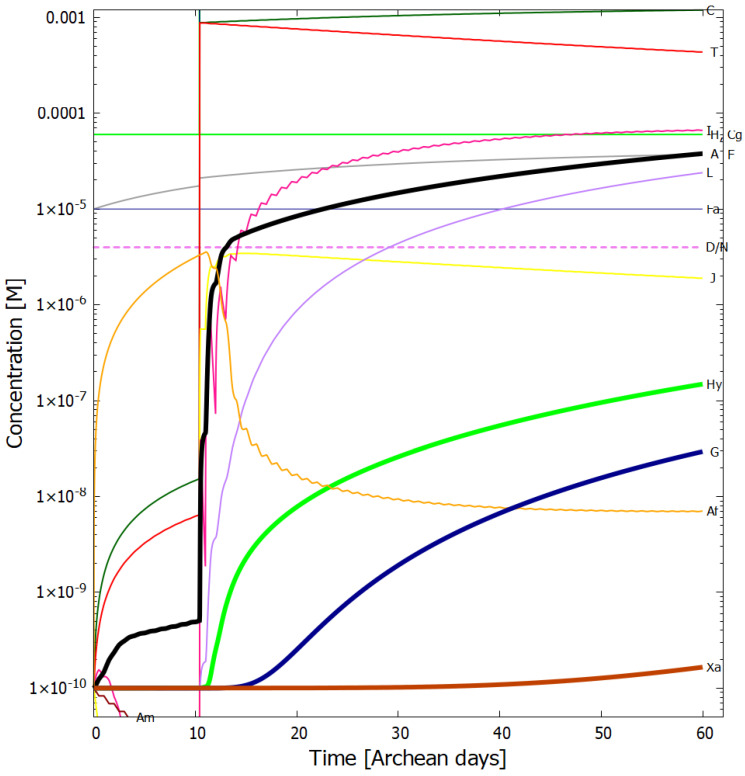
The same as for Figure 5 (i.e., temperature of 80 °C) but with 1/100 of the incident light flux (i.e., UV-C flux 1.25 W m${}^{-2}$
μm${}^{-1}$ at 260 nm, integrated flux 210-285 nm = 0.040 W m${}^{-2}$). This would correspond to the illumination on Earth’s surface if the only greenhouse gas to raise the surface temperature was CO${}_{2}$, assuming the standard solar model. The adenine concentration reaches $3.8\times 10^{-5}$ M and a guanine concentration of $2.9\times 10^{-8}$ M after 60 Archean days.

**Figure 7 entropy-24-01027-f007:**
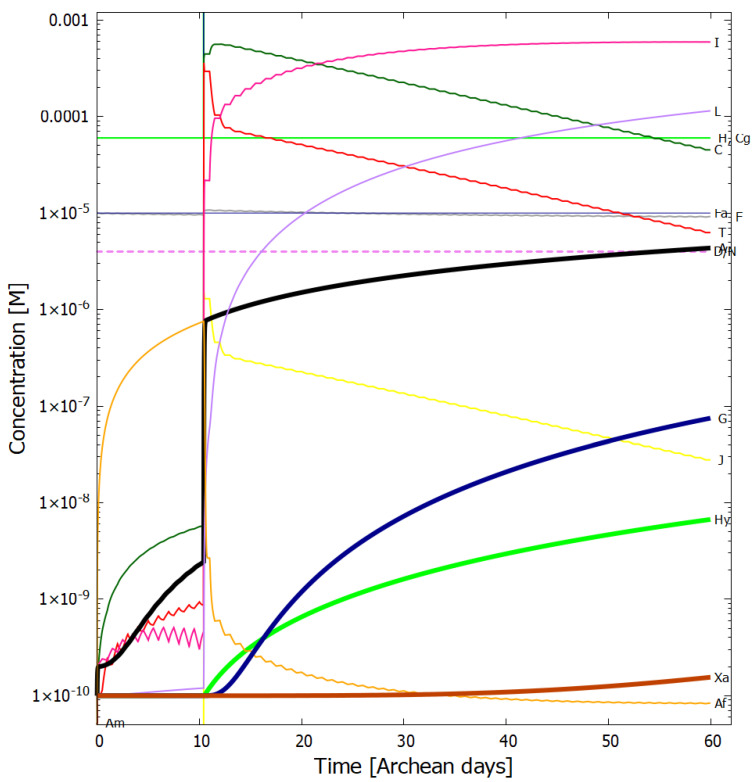
The same as for Figure 4 except for a temperature of 70 °C. The adenine concentration reaches $4.3\times 10^{-6}$ M and a guanine concentration of $7.5\times 10^{-8}$ M after 60 Archean days.

**Figure 8 entropy-24-01027-f008:**
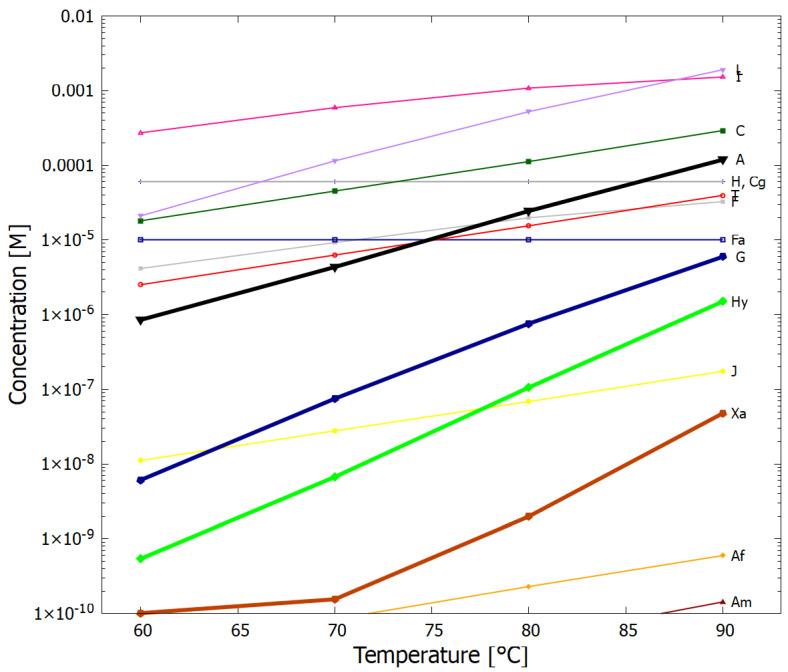
The temperature dependence of the concentrations of the product molecules after 60 Archean days, with the initial conditions; UV-C flux 125 W m${}^{-2}$
μm${}^{-1}$ at 260 nm (Figure 1), integrated flux 210–285 nm = 4 W m${}^{-2}$, [H]${}_{0}=6\times 10^{-5}$, [Cg]${}_{0}=6\times 10^{-5}$, [F]${}_{0}=1\times 10^{-5}$, [Fa]${}_{0}=1\times 10^{-5}$ M, and all other molecules [Y]${}_{0}=1\times 10^{-10}$ and the diffusion constant $D_{A}\left(300\right)=1.0\times 10^{-6}$, with one perturbation of [H], [Cg] and [Fa] to 0.1 M for 2 min at 10.4 Archean days.

**Figure 9 entropy-24-01027-f009:**
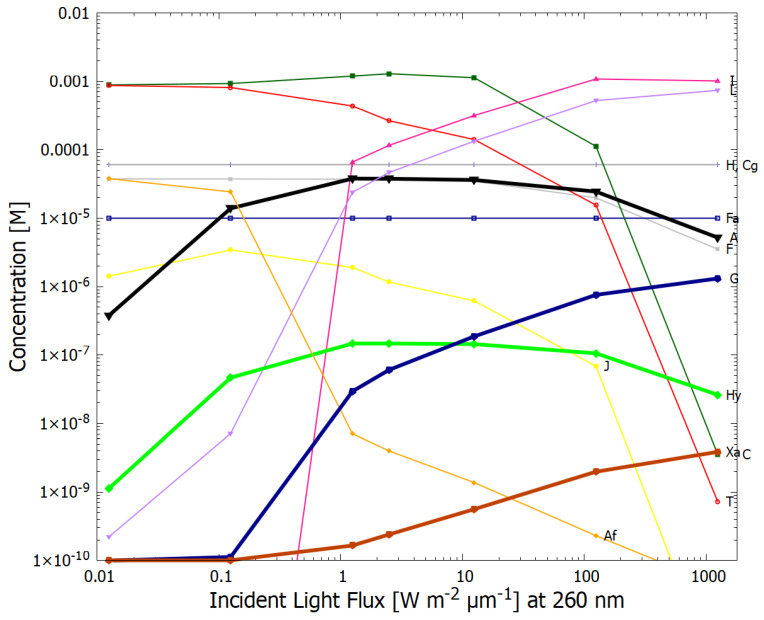
The dependence of the concentrations of the product molecules after 60 Archean days on the incident UV-C light intensity, with the initial conditions, T = 80 °C, [H]${}_{0}=6\times 10^{-5}$, [Cg]${}_{0}=6\times 10^{-5}$, [F]${}_{0}=1\times 10^{-5}$, [Fa]${}_{0}=1\times 10^{-5}$ M, and all other molecules [Y]${}_{0}=1\times 10^{-10}$ and the diffusion constant $D_{A}\left(300\right)=1.0\times 10^{-6}$, with one perturbation of [H], [Cg] and [Fa] to 0.1 M for 2 min at 10.4 Archean days.

**Figure 10 entropy-24-01027-f010:**
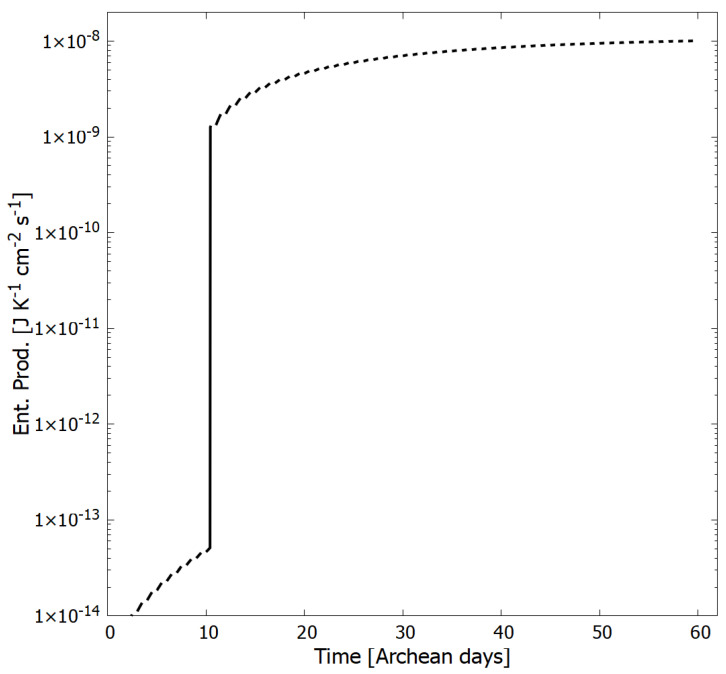
The entropy production as a function of Archean days for our UV-C photochemical dissipative structuring system within a vesicle floating at a sea surface at a temperature of 80 °C leading to the purines. Entropy production increases monotonically as photochemical reactions convert HCN into the photon dissipative product molecules, including adenine and guanine. During the day, entropy production is due to the dissipation of the UV-C spectrum into heat by the corresponding product concentration profile. During the night, entropy production is due to thermal chemical reactions, however, this entropy production is small and not included in the figure. At 10.4 Archean days, the system is perturbed and moves into a new stationary state with the entropy production increasing by more than 4 orders of magnitude and continues to increase as the concentration profile evolves (see Figure 5).

**Table 1 entropy-24-01027-t001:** Nomenclature, chemical formula, abbreviation (used in the text and in kinetic equations), label in Figure 3, wavelength of maximum absorption $\lambda_{max}$ (Figure 1), molar extinction coefficient at that wavelength $\epsilon_{max}$, electric dipole moment μ, and the topological polar surface area (TPSA), of the molecules involved in the photochemical synthesis of the purines. Values marked with “*” are estimates, obtained by comparing to similar molecules, for cases where no published data have been found.

Name	Chemical Formula	Abbrev. in Text	Abbrev. in Kinetics	Figure 3	$\lambda_{\max}$ nm	$\epsilon_{\max}$ M${}^{-1}$ cm${}^{-1}$	μ [D]	TPSA [Å${}^{2}$]
hydrogen cyanide	HCN	HCN	H	1			2.98	23.8
cyanogen	NCCN	NCCN	Cg	10			0.00	47.6
formamide	H${}_{2}$N-CHO	formamide	F		220	60 [55,56]	4.27 [57]	43.1
formimidic acid	H(OH)C=NH	formimidic acid (trans)	Fa		220	60	1.14 [57]	43.1 *
ammonium formate	NH${}_{4}$HCO${}_{2}$	ammonium formate	Af				+/−, 2.0 *	41.1
diaminomaleonitrile	C${}_{4}$H${}_{4}$N${}_{4}$	cis-DAMN (DAMN)	C	2	298	14,000 [58]	6.80 [59]	99.6
diaminofumaronitrile	C${}_{4}$H${}_{4}$N${}_{4}$	trans-DAMN (DAFN)	T	3	313	8500 [58]	1.49 [59]	99.6
2-amino-3-iminoacrylimidoyl cyanide	C${}_{4}$H${}_{4}$N${}_{4}$	AIAC	J	4	275	9000 [5,60]	1.49	99.6 *
4-aminoimidazole-5-carbonitrile	C${}_{4}$H${}_{4}$N${}_{4}$	AICN	I	7	250	10,700 [58]	3.67	78.5
4-aminoimidazole-5-carboxamide	C${}_{4}$H${}_{6}$N${}_{4}$O	AICA	L	10	266 [39]	10,700 *	3.67 *	97.8
5-(N’-formamidinyl)-1H-imidazole-4-carbonitrileamidine	C${}_{5}$H${}_{5}$N${}_{5}$	amidine	Am		250	10,700 [61]	6.83 *	80.5 *
adenine	C${}_{5}$H${}_{5}$N${}_{5}$	adenine	A	8	260	15,040 [62]	6.83 [63]	80.5
hypoxanthine	C${}_{5}$H${}_{4}$N${}_{4}$O	hypoxanthine	Hy	9	250	12,500 [64]	3.16	70.1
xanthine	C${}_{5}$H${}_{4}$N${}_{4}$O${}_{2}$	xanthine	Xa	12	268 [4]	9300 [2]	4.46 [6]	86.9 [11]
guanine	C${}_{5}$H${}_{5}$N${}_{5}$O	guanine	G	11	252 [62]	14,090 [62]	5.45 [65]	96.2 [65]

**Table 2 entropy-24-01027-t002:** Reactions involved in the photochemical synthesis of the purines (see Figure 3). Temperature *T* is in Kelvin and all kinetic parameters were obtained at pH 7.0.

#	Reaction	Reaction Constants
1	H $\overset{k_{1}}{⇀}$ F	$k_{1}=\exp\left(-14,039.0/T+24.732\right)$; s${}^{-1}$; hydrolysis of HCN [5,91,94]
2	$\gamma_{220}+$ F → Fa	$q_{2}=0.05$ [55,56,90,92,93]
3	$\gamma_{220}+$ Fa → H + H${}_{2}$O	$q_{3}=0.03$ [92,93,97]
4	F $\overset{k_{4}}{⇀}$ Af	$k_{4}=\exp\left(-13,587.0/T+23.735\right)$; s${}^{-1}$; hydrolysis of formamide [93,94]
5	4H $\overset{k_{5}}{⇀}$ C	$k_{5}=1/\left(\exp\left(-\Delta E/RT\right)+1\right)\cdot \exp\left(-10,822.37/T+19.049\right)$; M${}^{-1}$ s${}^{-1}$; ΔE=0.61 kcal mol${}^{-1}$ [5]
6	4H $\overset{k_{6}}{⇀}$ T	$k_{6}=1/\left(\exp\left(+\Delta E/RT\right)+1\right)\cdot \exp\left(-10,822.37/T+19.049\right)$; M${}^{-1}$ s${}^{-1}$; tetramization [5]
7	4H + T + Cg $\overset{k_{7}}{⇀}$ C + T + Cg	$k_{7}=\left(1.0/\left(1.0\cdot 0.01\right)\right)\exp\left(-\left(10,822.37-728.45\right)/T+19.049\right)$; M${}^{-2}$ s${}^{-1}$ [5]
8	4H + T + Cg $\overset{k_{8}}{⇀}$ 2T + Cg	$k_{8}=k_{7}$; M${}^{-2}$ s${}^{-1}$ [5]
9a	$\gamma_{298}$ + C → T	$q_{9}=0.045$ [58]
9b	$\gamma_{313}$ + T → C	$q_{9r}=0.020$ [5,58,60]
10	$\gamma_{313}$ + T → J	$q_{10}=0.006$ [5,58,60]
11	$\gamma_{275}$ + J → I	$q_{11}=0.583$; T →I; $q_{10}\times q_{11}=0.0034$ [5,60]
12	I $\overset{k_{12}}{⇀}$ L	$k_{12}=\exp\left(-E_{a}/RT+12.974\right)$; s${}^{-1}$; $E_{a}=19.93$ kcal mol${}^{-1}$; hydrolysis of AICN [6]
13	I:F + Af $\overset{k_{13}}{⇀}$ A + F	$k_{13}=\exp\left(-E_{a}/RT+12.973\right)$; M${}^{-1}$ s${}^{-1}$; $E_{a}=6.68$ kcal mol${}^{-1}$ [98,99]
14	I:F + Fa $\overset{k_{14}}{⇀}$ Am + Fa +H${}_{2}$O	$k_{14}=\exp\left(-E_{a}/RT+12.613\right)$; M${}^{-1}$ s${}^{-1}$; $E_{a}=19.90$ kcal mol${}^{-1}$ [100]
15	$\gamma_{250}$ + Am → A	$q_{15}=0.060$ [61]
16	A $\overset{k_{16}}{⇀}$ Hy	$k_{16}=10^{\left(-5902/T+8.15\right)}$; s${}^{-1}$; valid for pH within 5 to 8; hydrolysis of adenine [89,95]
17	L + Cg $\overset{k_{17}}{⇀}$ G + H	$k_{17}=\exp\left(-E_{a}/RT+15.52\right)$; M${}^{-1}$s${}^{-1}$; AICA + Cyanogen, $E_{a}=18.49$ kcal mol${}^{-1}$ [6]
18	G $\overset{k_{18}}{⇀}$ Xa	$k_{18}=10^{\left(-6330/T+9.40\right)}$; s${}^{-1}$; valid for pH within 5 to 8; hydrolysis of guanine [89,95]
19	$\gamma_{298}$ + C → C	$q_{19}=0.955$
20	$\gamma_{313}$ + T → T	$q_{20}=0.972$
21	$\gamma_{275}$ + J → J	$q_{21}=0.417$
22	$\gamma_{250}$ + Am → Am	$q_{22}=0.940$
23	$\gamma_{250}$ + I → I	$q_{23}=1.000$
24	$\gamma_{266}$ + L → L	$q_{24}=1.000$
25	$\gamma_{260}$ + A → A	$q_{25}=1.000$
26	$\gamma_{250}$ + Hy → Hy	$q_{26}=1.000$
27	$\gamma_{252}$ + G → G	$q_{27}=1.000$
28	$\gamma_{268}$ + Xa → Xa	$q_{28}=1.000$

**Table 3 entropy-24-01027-t003:** Diffusion constants at 80 °C (353.15 K) relative to that of adenine at 300 K (i.e., $d_{Y}=D_{Y}\left(353.15\right)/D_{A}\left(300\right)$ obtained from Equation (Equation 18). In this paper, $D_{A}\left(300\right)$ is taken to be $1\times 10^{-6}$ cm${}^{2}$ s${}^{-1}$.

$d_{H}$	$d_{F}$	$d_{\mathrm{Fa}}$	$d_{\mathrm{Af}}$	$d_{C}$	$d_{T}$	$d_{J}$	$d_{I}$	$d_{L}$	$d_{\mathrm{Am}}$	$d_{A}$	$d_{\mathrm{Hy}}$	$d_{\mathrm{Cg}}$	$d_{G}$	$d_{\mathrm{Xa}}$
3.84	2.26	4.52	3.64	1.11	2.81	2.81	1.86	1.66	1.18	1.18	2.16	6.76	1.28	1.55

## Data Availability

Not applicable.

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
