# Peer review of "Dissipative Photochemical Abiogenesis of the Purines"

_entropy, 2022, doi:10.3390/e24081027_

Round 1
Reviewer 1 Report
Reviewer’s Report: The Dissipative Photochemical Origin of Life: UVC Abiogenisis of the Purines, by Claudeth Hernández and Karo Michaelian.
General Comments: The manuscript presents a study focused on the role of ultraviolet (UV) light regarding the abiogenic origin of life. Briefly, the authors have developed a model of the Archean in which the interaction between (CN)2, HCN, H2O, and UV radiation led to higher concentrations of some of the nucleobases than what is predicted by previous models for primeval Earth.
The manuscript is very interesting and well written, even though there are a few points which I believe should be addressed before it can be published. In my opinion, the main issue of this study is that the flux of UV light on Earth’s surface could have been severely overestimated by the authors. The work of Michaelian 2022 is used as reference for the UV flux during the Archean which, if integrated between 200 and 280 nm range, yields to a flux of ~5.1 W m-2. However, other sources provide vastly different values, for example, Doglioni et al. (2016 - Geoscience Frontiers, 7(6), 865-873), estimated an integrated flux of only 0.30 W m^-2 for the same period and range (the integration was done by me based on published data). Since the flux of UVC is one of the most important aspects of their model, I would like to see stronger evidence that the flux of UV light considered in their model is feasible. In the same sense, when the authors say “[HCN] production during the Hadean and Archean on Earth was probably a result of the solar Lyman alpha line (121.6 nm) photo-lysing N2 in the upper atmosphere” (lines 84-86) I am not sure if this possibility is feasible since Ly-alpha is readily absorbed by oxygen at the highest layers of atmosphere. For this reason, I would like to see a reference for this statement. The same comment is also valid for what is written in lines 278-279: “HCN can also be produced through the solar Lyman alpha line (121.6 nm) photo-lysing N2 in the upper atmosphere”. To be clear, the issue I see is not with the photolysis of N2 in itself, but with the feasibility of it happening due to Ly-alpha in the upper atmosphere – I am not certain that Ly-alpha can penetrate deep enough to interact with N2.
- Another general issue I find is concerning the "Discussion" (line 382) and "Conclusions" (line 402) sections. The Discussion begins “too abruptly”, i.e., there is no contextualization. The authors simply begin the discussion by stating that “High temperatures lead to a faster buildup of product concentration (Fig. 6) and high temperatures...” and so on. Not mentioning the fact that this section is rather short. I think that the "Discussion" section must be rewritten. The "Conclusions" section is also somewhat short and could be improved. The results of this work are very interesting, but this is not reflected in the "Conclusions" section in my opinion.
Specific comments:
- Title, Abstract: The most common spelling I could find is “abiogenesis”, not “abiogenisis”
- Line 2: Please define “UVC pigments”
- Line 23: please name the HCN molecule
- Line 23: “hydrolysis” is misspelled
- Line 24: Please write the formulas of formamide, amonium formate, cyanogen and cyanate
- Line 35: please define UVC
- Line 49: “somewhere” is misspelled
- Line 52: “perhaps” is misspelled
- Line 89: Only here the formula of cyanogen is mentioned for the first time in the manuscript. Please provide the chemical formula of cyanogen when this molecule is mentioned for the first time in the manuscript.
- Lines 366-367: as mentioned above, different authors provide other figures for the flux of 200-280 nm photons at Earth’s surface during the Archean. According to Doglioni et al. 2016, for example, the integrated flux of UV light in this range at surface level during the Archean was only ~0.30 W m^-2.
- Line 371: “reaction” is misspelled
-Figure 6: some of the characters are difficult to read, but this issue can be fixed by the editing team later.
Reviewer 2 Report
This study, a continuation of other studies, models the origin of life as a photochemical dissipative structuring process with reaction-diffusion occurring within a fatty acid vesicle floating on a hot ocean surface, permeable to the precursor molecules HCN, cyanogen, and H2O, and under a UVC (210-285 nm) and UVA (310-360 nm) photon flux. The study details those most important aspects, of the photochemistry involved in the non-equilibrium thermodynamics of dissipative structuring and proliferation, pertaining to the dissipative structuring of the purines guanine and xanthine.
For the developed models to be realistic the following points, related to fatty acid vesicle stability, should be also addressed:
1. The temp of the hot ocean surface the fatty acid vesicle are floating on.
2. The kind/s of primitive fatty acids needed to form stable vesicles in a hot ocean surface, in reference to carbon number length, saturation/unsaturation, branching; e.g. low carbon number length destabilises fatty acid vesicles even at 25C, while saturated fatty acids make more stable vesicles at high temp. The authors are advised to look at the following indicative studies:
Georgiou, C. D., Deamer, D. W. (2014). Lipids as universal biomarkers of extraterrestrial life. Astrobiology 14(6): 541-549
Deamer, D. W., Georgiou, C. D. (2015). Hydrothermal conditions and the origin of cellular life. Astrobiology 15: 1091-1095
3. The stability of primitive fatty acids, and their vesicles, under UVC/UVA (210-285 nm/310-360 nm) irradiation (combined with high ocean temp), which causes oxidative damage in the structure of fatty acids, especially the unsaturated ones; see Halliwell, B., Gutteridge, J.M.C. (2015). Free radicals in biology and medicine. 5th ed., Oxford University Press, Oxford
4. The salinity of the oceans: high salinity destabilises fatty acid vesicles even at 25C (see mentioned study by Deamer et al. 2015)
